# Development of a Smoke-Free Home Intervention for Families of Babies Admitted to Neonatal Intensive Care

**DOI:** 10.3390/ijerph19063670

**Published:** 2022-03-19

**Authors:** Caitlin Notley, Tracey J. Brown, Linda Bauld, Elaine M. Boyle, Paul Clarke, Wendy Hardeman, Richard Holland, Marie Hubbard, Felix Naughton, Amy Nichols, Sophie Orton, Michael Ussher, Emma Ward

**Affiliations:** 1Norwich Medical School, University of East Anglia, Norwich NR4 7TJ, UK; tracey.j.brown@uea.ac.uk (T.J.B.); paul.clarke@nnuh.nhs.uk (P.C.); emma.ward@uea.ac.uk (E.W.); 2Usher Institute and SPECTRUM Consortium, College of Medicine and Veterinary Medicine, University of Edinburgh, Edinburgh EH8 9AG, UK; linda.bauld@ed.ac.uk; 3Department of Health Sciences, University of Leicester, Leicester LE1 7RH, UK; eb124@leicester.ac.uk; 4Neonatal Unit, Leicester Royal Infirmary, Leicester LE5 4PW, UK; marie.hubbard@uhl-tr.nhs.uk; 5Neonatal Intensive Care Unit, Norfolk and Norwich University Hospitals NHS Foundation Trust, Norwich NR4 7UY, UK; amy.nichols@nnuh.nhs.uk; 6School of Health Sciences, University of East Anglia, Norwich NR4 7TJ, UK; w.hardeman@uea.ac.uk (W.H.); f.naughton@uea.ac.uk (F.N.); 7Leicester Medical School, University of Leicester, Leicester LE1 7HA, UK; rch23@leicester.ac.uk; 8Division of Primary Care, University of Nottingham, Nottingham NG7 2RD, UK; sophie.orton@nottingham.ac.uk; 9Population Health Research Institute, St George’s, University of London, London SW17 0RE, UK; mussher@sgul.ac.uk; 10Institute for Social Marketing and Health, University of Stirling, Stirling FK9 4LA, UK

**Keywords:** neonatal, smoking cessation, smoke-free homes, relapse prevention, intervention development

## Abstract

Neonatal intensive care units (NICUs) have a disproportionately higher number of parents who smoke tobacco compared to the general population. A baby’s NICU admission offers a unique time to prompt behaviour change, and to emphasise the dangerous health risks of environmental tobacco smoke exposure to vulnerable infants. We sought to explore the views of mothers, fathers, wider family members, and healthcare professionals to develop an intervention to promote smoke-free homes, delivered on NICU. This article reports findings of a qualitative interview and focus group study with parents whose infants were in NICU (n = 42) and NICU healthcare professionals (n = 23). Thematic analysis was conducted to deductively explore aspects of intervention development including initiation, timing, components and delivery. Analysis of inductively occurring themes was also undertaken. Findings demonstrated that both parents and healthcare professionals supported the need for intervention. They felt it should be positioned around the promotion of smoke-free homes, but to achieve that end goal might incorporate direct cessation support during the NICU stay, support to stay smoke free (relapse prevention), and support and guidance for discussing smoking with family and household visitors. Qualitative analysis mapped well to an intervention based around the ‘3As’ approach (ask, advise, act). This informed a logic model and intervention pathway.

## 1. Introduction

Tobacco smoking has a severely detrimental impact on parental and child health [1,2]. Pregnant women who smoke are more likely to give birth to low birthweight babies [3] and to suffer premature births [4]. Infants born preterm or low birthweight babies are likely to need additional care and support in early life and many require admission to a neonatal intensive care unit (NICU), often for many weeks and months. The relative risk of admission to a NICU for infants of women who smoke is increased by at least 20% compared to infants of non-smoking mothers [5], and infants born to parents who smoke are likely to need a longer NICU stay [6] (6. Furthermore, a NICU admission results in significant costs to the healthcare system [7].

Infants born preterm often need critical care. Lung development, in particular, is incomplete, and many babies need significant medical intervention and protracted respiratory support. These infants are vulnerable to infection. Children are more susceptible to second-hand or environmental tobacco smoke (ETS) than are adults, particularly vulnerable children, such as infants born preterm [8]. It is known that direct exposure to ETS from parents or caregivers increases rates of sudden infant death syndrome, respiratory conditions, and other infections [9]. ETS exposure ‘may potentially hasten, delay, or prevent resolution of lung injury in preterm children’ [10]. Indirect exposure may also be a risk, and maternal smoking increases the odds twofold for developing chronic lung disease after preterm birth [11]. Studies have found smoking particles on NICU furniture and incubators [8]. Early exploratory evidence also suggest that ‘third-hand smoke exposure’ (to particles) was associated with microbiome differences in NICU-admitted infants [12]. Exposure to environmental toxins and allergens, including carcinogens from tobacco smoke, does not yet have proven direct links to adverse outcomes, although this exploratory evidence suggests there is a clear theoretical link that such exposure is likely to be damaging [13].

Smoking prevalence of parents of babies admitted to a NICU is higher than in the general population. In development work, our team found that approximately a third of parents asked on admission were current smokers, and approximately a third were recent ex-smokers [14], compared to the current UK population level smoking prevalence of less than 15% [15]. The NICU admission is an extremely stressful and anxiety-provoking situation for new parents, meaning that recent ex-smokers may be liable to relapse to tobacco smoking at this difficult time, as stress is a major predictor of smoking relapse postpartum [16]. Smoking prevalence is also higher in lower socioeconomic groups, suggesting marked health inequalities [17].

The birth of a child, particularly where the birth is preterm, offers a ‘teachable moment’ to support parents to quit smoking, remain smoke-free, and maintain smoke-free environments [18]. Parents can feel extremely helpless following the birth of a baby requiring NICU admission [19]. The preterm birth of an infant and the subsequent time spent in the NICU are extremely stressful for mothers and fathers, and may cause enduring stress symptoms lasting many years [20,21]. Aside from breastfeeding, stopping smoking, and staying smoke-free, is the most important positive behaviour that parents who smoke have control over that will likely impact on the morbidity outcomes for their baby, both immediately, and in terms of longer-term development throughout childhood.

National guidance recommends support for smoke-free strategies in secondary care settings during pregnancy and after childbirth [22,23,24]. Recent policy also emphasizes a focus on pregnancy and the post-partum period as key to meet the UK challenge of reducing rates of smoking to meet an ambitious target of a ‘smoke-free 2030’ [25]. However, interventions to maintain smoke-free environments are not routinely offered in UK NICUs [26,27]. Interventions trialed in this setting in the USA have included motivational interviewing and the offer of incentives, demonstrating some promise [28]. A recent review of interventions to prevent ETS in paediatric settings concluded that interventions should incorporate effective behaviour change techniques (BCTs) [29], suggesting the need to develop a tailored intervention to support families in NICUs specifically. Behaviour change interventions are complex by nature, comprising multiple components including different mechanisms of delivery in addition to BCTs [30]. In response, this study sought to understand the wants, needs, and experiences of parents who smoke who have a baby admitted to a UK NICU, alongside the views of other family members and health care professionals (HCPs) within NICUs. Following MRC guidance [31], we used qualitative methods to develop an intervention, which is relevant to the population it targets, is acceptable, feasible to implement and, therefore, likely to be effective, cost effective, and sustainable for promoting smoke-free home environments.

## 2. Materials and Methods

This qualitative study took a theory-based approach to intervention development, underpinned by a logic model that was adapted throughout the study (Figure 1). This was derived from existing evidence and qualitative developmental work informing the current study. The intervention model was driven by the capability–opportunity–motivation (COM-B) model of health behaviour change [22]. We used the COM_B model to help identify key potential mechanisms of action and then specified BCTs which would likely target these to ultimately map to the outcome of positive behaviour change (implementation of an intervention to support families to maintain a smoke-free home).

This study was originally planned as a qualitative focus group study with parents (who smoked or family members who were interested in discussing smoke free homes) of babies admitted to a NICU, to gain feedback on potential intervention components and to discuss issues of tailoring, timing, and intensity. Full ethical approval (REC: 19/EM/0235) allowed flexibility to also conduct one-to-one interviews with parents if this was their choice. On commencement, it quickly became apparent that focus groups were not feasible to organise due to time constraints of parents on NICU, or preferable, as most parents stated that they would prefer a couple or one to one interview; thus, we switched toward offering individual or dyad interviews for parents, and focus groups for health care professionals.

We used examples of prototype intervention components as discussion prompts (e.g., asking about timing of intervention, examples of modes of delivery, such as leaflet, website, app), in order to gather qualitative data to enable us to refine and tailor the intervention. We also explored views on procedures for optimising intervention delivery: who will initially deliver support on the unit; frequency and timing of support; and follow-up post discharge; as well as individual tailoring by considering factors, such as age, social, and cultural differences.

We recruited from two study sites to ensure maximum variation of population characteristics, such as age, socioeconomic status, ethnicity and severity of neonatal illness. Norfolk and Norwich University Hospitals NHS Foundation Trust (NNUH) NICU is the site where the lead research nurse had previously undertaken PPI activity. This NICU is one of three regional NICU centres in a Neonatal Operational Delivery Network serving the East of England. It undertakes a wide range of neonatal intensive care and is a highly research active unit. In 2018, the Neonatal Intensive Care Unit cared for 1170 infants. University Hospitals of Leicester NHS Trust (UHL) Neonatal Service is one of the largest units in the country and is a highly research active unit serving an ethnically and socioeconomically diverse patient population.

Eligibility criteria for recruitment to the qualitative study were:
**Inclusion**
Parent or family member—aged 16 years or over—of a baby currently admitted to NICU for a minimum of 24 h.Has capacity to give informed consent.
**Exclusion**
Parents/family members who lack capacity to consent.Insufficient fluency in English.Parents who have a baby admitted to NICU for less than 24 h.Parents for whom clinical judgement suggests it is not appropriate to discuss smoking cessation with (e.g., critical illness of infant).

Once initial interest was confirmed, the study was discussed with participants by the assigned neonatal nurse, face-to-face on the NICU at a convenient and sensitively chosen time. It was carefully explained to participants, and reiterated, that they did not have to answer any questions during data collection that they did not wish to, and that they were free to withdraw at any time, without giving a reason. Potential participants were given the opportunity to ask questions, and if they were interested in taking part, were handed the participant information sheet and a consent form. All of those approached were given at least 24 h to decide whether they wished to participate, prior to providing informed consent. Following consent, the research nurse assigned an anonymised participant study number to the participant, and they were then asked to complete a short demographic questionnaire. The interview followed a semi-structured topic guide (Appendix A) and we showed prompts to give examples of potential intervention components to aid discussion. In total, we consented 44 parents, of whom, 42 participated in an interview. We removed the details and did not contact (for an interview) a mother and a father whose baby later died. All participants taking part in interviews received a GBP 20 voucher in acknowledgement of their time.

Originally, we estimated that we would need to undertake approximately ten focus groups with approximately six participants per group, maximising opportunities for discussion and consensus forming. As we changed our approach to collecting data via individual or dyad interview as an alternative, our final sample size of 42 was deemed sufficient to have reached saturation of themes emerging in the data [32].

### Health Care Professionals Focus Groups

Health professionals (HCPs) working with families of babies admitted to a NICU, or smoking cessation professionals were identified and invited to take part in focus groups, or one-to-one interviews (in-person or by telephone). We had initially planned to recruit approximately 12 NICU health care professionals (e.g., neonatal nurses, doctors). On completion of this phase, we consented and recorded the views of a total of 23 HCPs via remote (video conferencing) focus groups.

Analysis of qualitative data took a combined deductive/inductive thematic coding approach [33]. We deductively coded responses around intervention delivery following our topic guide, and inductively coded issues naturally arising during interviews and focus groups. The analysis was conducted by one researcher (either TB or EW) with secondary coding and consensus of analysis agreed by the lead author (CN).

Qualitative findings are reported summarising the thematic analysis. Data extracts are provided as best illustrations of the themes presented. Participants are anonymised and coded as *S* for smoker, *ES* for ex-smoker, *NS* for non-smoker; *NNUH (Norwich)* or *UHL (Leicester)* define the location.

## 3. Results

We included in the analysis 42 parents in total (Table 1)—22 mothers, 18 fathers, 1 partner (unrelated to the baby), and 1 grandparent. Our sample included some ethnic diversity, approximately reflecting the local populations served by the participating NICUs, including one person of mixed ethnic origin, seven people of Asian origin, and one Black/African/Caribbean participant. There was also a representative spread of educational attainment across the sample, with 5 people having no formal qualifications, 10 with GCSEs or equivalent, 11 with A levels or equivalent, and 8 educated to degree level or above. For smoking status, we recruited a mix of smokers and non-smokers—our sample included 10 active current daily tobacco smokers, 2 recent ex-smokers, 6 long-term ex-smokers, and 24 people who had never smoked or only experimented with tobacco but never regularly smoked. We had just one current e-cigarette user in the sample. Of the 10 current smokers in our sample, 5 (50%) said they would consider quitting, 3 (30%) said that they were actively trying to stop smoking, and 2 (20%) stated that they were not interested in stopping smoking.

We gained views and feedback on intervention components, timing, and delivery parameters from 23 nurses and senior nurses, consultant neonatologists, junior doctors, and health visitors (Table 2).

Qualitative data followed our topic guide, exploring various elements of a potential intervention approach, organised below, thematically, from our analysis, which identified key themes.

## 4. Findings

### 4.1. Intervention Suitability and Culture Change

Having a baby admitted to a NICU was considered a turning point, a potential ‘teachable moment’. The immediacy of the emergency admission, often completely unexpected, meant that the health of their baby became paramount to parents/family members. There was enormous respect for the clinicians caring for the baby, and a strong desire to do anything possible to be able to positively influence outcomes. Parents demonstrated how the situation made them more receptive to making personal changes regarding their own smoking:


*“When you’re 25 and you’re told you might die when you’re 80 you’re like, ‘might get hit by a bus tomorrow so I’m really not that bothered’. But if you find out that you might kill your baby in 6 months’ time, you’re giving up tomorrow.”*
(S LL03-05)

Some parents were worried about potential exposure of smoking particles, either through exposure on the NICU from other visitors, or from family members once they returned home:


*“You do have to think about the health of your baby and other people’s because, there’re parents up here that don’t even smoke and you’re on the ward and you’re walking around, you’re walking past their babies and stuff like that.”*
(S NNUH16)

Parents themselves suggested that NICU admission for their baby was an important and relevant opportunity for smokers to quit and expressed surprise that the topic of tobacco smoking was not routinely raised:


*“Definitely because when you’re in NICU they give you all the information about how to save your baby’s life on discharge and safe sleeping but at no point do they say, ‘do you smoke?’ are you going to be around your baby whilst smoking?’ and I think that’s probably quite important.”*
(ES UHL 009-010)

As with many parents, when asked, HCPs recognised the need for intervention. They highlighted that the baby was their patient and that they should be advocating for them to optimise the environment for the baby. Many drew parallels to breastfeeding and believed that having conversations about smoke-free environments with all parents (including non-smokers), and offering cessation support to those that smoked, should become routine practice, helping to normalise the message:


*“With parents and smoking, if there’s evidence to suggest actually long term the baby’s going to do far better if you’ve got a non-smoker and then a non-smoke house then if you have that conversation right at the beginning, as part of what we all plan to do, which is basically give them their baby in the best possible condition they can with the least long-term issues that we can, that can almost be seen as part of the norm.”*
(FG 24.09)

Both HCPs and parents strongly agreed that both mothers and their partners should be included in discussions about smoke-free households and be offered smoking cessation support if needed. Some fathers also expressed that they felt it would empower them to be involved as they often felt left out of maternity and postnatal care. Parents felt that it would be easier to quit if they were both attempting it, as they could support each other.


*“The mums need the dads’ support and if they [the dads] perhaps gave up whilst the mum’s pregnant like with the mum, then perhaps the mum wouldn’t be so inclined to go back to smoking.”*
(S NNUH16)

#### Barriers

The biggest concern both parents and HCPs had about the suitability of intervening in NICU was that it could shame parents, making them feel they were to blame for their baby’s health:


*“If I was a smoker, if I had spent my pregnancy smoking, I’d already feel quite fragile knowing that my baby’s poorly enough to be in the neonatal unit and to be asked straight away ‘do you smoke?’ I’d probably feel: are they wondering if it’s my fault?”*
(Ex-smoker UHL 009-10)

HCPs were acutely aware of stigma around smoking, especially as there was often added complexity, as those parents who were smokers sometimes had other social and health issues that were also stigmatised. Smoking was seen as a ‘taboo’ subject and HCPs perceived that any conversation around it had the potential to undermine the relationship between HCPs and parents:


*“The people who smoke are mostly the poorest and the most disenfranchised and with a huge amount of mental health… mothers we have and substance misusers… when I visit people at home, I’m just grateful if they are putting on a different jacket to go outside their flat or dad is or granny. So we’re not talking to the middle classes, we’re talking to a lot of socioeconomic factors that are very, very hard to break down and if you start having too high standards, you’re actually making people with very low self-esteem feel even worse than they do and then you just don’t build any relationship.”*
(FG 24.09)

Although this was recognised as a barrier, it was not seen by some as a reason not to intervene, as there was a sense of duty and responsibility to have the conversation, no matter how difficult:


*“But then you’ve done your bit haven’t you. You’ve not passed the buck, but you’ve fulfilled your role by saying… you’ve asked them haven’t you. If they then choose not to answer, you can’t do nothing about that. It’s out of your hands then.”*
(S NNUH10-11)

However, HCPs commented that they lacked confidence to start the conversation and would welcome an intervention providing them with a framework to talk to parents about this taboo subject in a way that did not stigmatise parents at this vulnerable time:


*“I think I mean, yeah, it’s conversations that if we know that they’re a newly ex-smoker or you know, we know that they’re going through a stressful point. It’s about us having the tools and the right language and the right teaching to speak to them about it.”*
(FG 18.09)

### 4.2. Intervention Delivery

Our topic guide explored timing and pragmatic aspects of intervention delivery, asking parents their views on what might be needed, what would be acceptable, and particularly also exploring with HCPs what might be feasible to implement.

#### 4.2.1. Intervention Timing

Parents and HCPs thought there should be a flexible approach to initiating discussion with parents about the impacts of smoking and available help. For some, it might be appropriate to offer help right from admission in order to intervene before smoking habits had become ingrained on visits to the unit, whereas for others it would be better to wait at least a day before broaching the subject. HCPs were concerned that parents were overloaded with information upon admission and would not have capacity to process the information due to the stress of the situation, as one parents said:


*“For someone to then come up to me and go ‘do you want help to stop smoking?’. I don’t even know how I’m standing on my feet and I how I’m managing to walk in a straight line right now and how I’m getting out of bed.”*
(S NNUH05-04)

Doctors were seen as credible, prompting parents to listen, and nurses were perceived to have a close relationship with the parents and more time to spend to discuss smoke-free environments:


*“Having a nurse who is looking after my child, saying in the first couple of days: ‘Look, I don’t mean to pry but are you a smoker? Because smoking has a very big impact on your child’s development and their lungs’. It doesn’t have to be a detailed explanation.”*
(NS NNUH02)

There was a consensus that discussion around smoke-free environments should not stop after this initial discussion but should be further reinforced at subsequent opportunities during the NICU stay. The Family Care Team, whose role it is to support the family throughout and beyond the NICU admission and, therefore, often had a strong relationship with the family, were mentioned by a few of the parents as being suitable to offer continued support around smoke-free environments:


*“I think maybe the family care team would be helpful to do it because they’re involved in like supporting the family. And that’s what it’s all about, looking after yourselves, your other children and your baby when they come home. So I think maybe someone like that.”*
(S NNUH16)

Parents commented that support around staying smoke free should continue after leaving the NICU. They recognised that this was a potential risky period for relapse:


*“Follow-up would be important because it’s very easy for somebody to give up smoking during that period, then the baby gets stronger, the baby comes home and they maybe think ok we’re out of the danger period now, I’ll just nip and have one and you’re back to a non-smoke free environment.”*
(ES NNUH 009-10)

Health visitors were thought to be ideally placed for follow-up once the baby had gone home as they were regularly seeing families. They could offer continuity of care handed over from the NICU outreach team and include smoke-free support in their existing scheduled meetings with families:


*“As health visitors we are quite skilled in having those conversations and having difficult conversations with parents.”*
(HV 14.10)

Continuity of care and support was felt to be vitally important, not just for mothers—who could struggle following discharge and feel slightly abandoned, but also for fathers—who play a vital role in supporting mothers and who have a large influence on maternal smoking status:


*“I just genuinely think that when the baby is discharged and [we] go home as a family, I really do think that extra support, not just for your little one but yourself as well because mums do feel rubbish when they go home. I really think mum as well as dad because dads you know they feel down sometimes.”*
(S NNUH 003-004)

#### 4.2.2. Training and Specialist Support

Basic training was discussed by participants as a way to enable staff to have conversations around smoke-free environments and discuss options for smokers:


*“So having everyone on the unit trained to a certain level of knowledge about smoking cessation and the health benefits of quitting. The health benefits of taking the baby home to a smoke-free environment.”*
(FG 13.10)

Some of the parents also liked the idea of a staff member who was a smoking cessation specialist being available if they needed further support around quitting smoking. HCPs drew parallels with the model of breastfeeding support and thought that having a dedicated champion would work well as it would offer staff support:


*“I like the idea of having someone with more knowledge because parents will always ask you questions that you don’t quite remember or you know that you’re not as familiar with that, the evidence on that, and it’s always good to have someone to signpost them to, to say ah well I can arrange for you to talk to such and such who has a lot more information on it. So I do like that idea.”*
(FG 13.01)

#### 4.2.3. Educational Resources

HCPs and parents discussed education materials, such as posters and leaflets as a way to deliver the intervention. They believed that these could be effective because they were accessible and because they could introduce or reinforce messages that could be followed up in a face-to-face discussion. Parents commented that they would flick through leaflets they had been given, or read posters on the wall, often to give them something to do:


*“It’s quite nice to sit and just focus and let your mind focus on that and the poster and you actually you’ll be surprised when I went home how much I actually remember from them posters.”*
(ES NS NNUH 003-04)

HCPs felt that it would be quite straightforward to include leaflets and posters around the unit and include leaflets in packs that they already gave out to families upon baby’s admission:


*“Yes, yes, I give everybody a … well my team, we give everybody an admission pack which has got lots of leaflets about safe sleep and resus [resuscitation] and all that kind of thing. So it definitely could go in there.”*
(FG 28.09)

Digital support in the form of apps, videos, websites, or text messages was discussed. Participants felt that digital support could be useful to reinforce the other support offered. Both groups agreed that it was a normative and accessible medium:


*“The app’s on your phone and you’re looking on your phone all the time. You might be like ‘ooh I’ve got a minute I’ll look through that now’. Do you know what I mean? Whereas the leaflet’s gone and forgotten but the app’s always there on your phone and you’ve got the reminder every time you go on your phone and it’s there to look at.”*
(ES UHL 006)

### 4.3. Intervention Components

Specific examples of intervention components were discussed during interviews. Components that parents felt would be helpful, supportive and acceptable were organized around a brief intervention framework (‘Ask, Advise, Act’).

#### 4.3.1. ASK—Identifying Babies at Risk of Tobacco Smoke Exposure

Currently, there is no formal process for identifying babies admitted to the NICU at risk of smoke exposure. The information is not guaranteed to be on the mother’s record from the booking appointment and, even if it is, it might not be up to date and will not include other household members’ smoking status or whether friends and family smoke. HCPs felt that aquestionnaire could be part of a universal lifestyle questionnaire administered to all NICU parents. They felt that standardising the approach could avoid the stigma of asking people directly. Parents and HCPs felt that a short questionnaire would be acceptable, as long as anonymity was assured:


*“I think maybe like a questionnaire or something like that. Because I think some parents might find it a little bit, I don’t know, overwhelming to have their baby in here and then to be asked ‘well do you smoke?’ kind of thing. They might find it a bit ooh!”*
(S NNUH16)

Carbon monoxide (CO) monitoring was also discussed as a possible way of identifying smokers entering the unit. Participants’ views on this were mixed. Some parents were interested in finding out their CO reading and the potential impact on the baby, which could prompt engagement with the intervention:


*“Personally, as a father rather than mother who’s gone through labour and childbirth and all the rest of it, if a doctor asked me to do a CO test to try and help my premature baby, see if anything I was doing would adversely affect it, then I’d be willing to do that, no problems, no issues … You want the best for your child, and you want it to be as easy as possible for them. If there’s anything that I was doing that could hinder recovery and growth and everything like that, I’d want to know how to do it.”*
(ES UHL 009-010)


*“We tend to find that CO testing is a really good way of mothers understanding how smoking is affecting their health and their babies’ health, but it also gives the health care provider a good opportunity to approach that subject with them as well by discussing carbon monoxide, so it’s a good all-round tool.”*
(FG 28.09)

Other participants, especially smokers, believed that CO testing upon admission would be too intrusive and potentially stigmatising:


*“I could see if you, straight off the bat, said ‘we test everyone as a minimum’. It almost assumes guilt and I think it gives that (impression) of we are, you know, we’re against you. I think some parents could feel like that.”*
(S & NS NNUH 001-002)

#### 4.3.2. ADVISE—Inform Parents about the Risk to the Baby’s Health

Parents wanted factual, hard hitting (meaning, factual messages should not be ‘softened’ due to causing potential distress), information about how their baby specifically would be affected by second and third-hand smoke (smoking particles). They acknowledged that although they knew the risks to themselves of smoking, they were not clear on the impact of smoking on the baby. They felt that knowing the risks could be very motivating in stopping smoking or ensuring a smoke-free environment:

“… if you now sat in front of me and said: ‘It’s harming the baby for you to be sitting next to her after you’ve had a fag’.
*NNUH05: Especially after everything they’ve been through.*

*NNUH04: I’d give up. I’ve watched that boy fight for the last 26 days and I’d feel disgusted to think I was making his life worse because I’d had a cigarette. (S NNUH 05-04)*

*If you give it to a lot of other parents as well and said, ‘you smoking and having it on your clothes is harming your kids and that is an absolute, 100% fact that that is harming kids and you yourself’. Because it’s like saying I’m sitting here right now and I’m hurting myself.”*
(S NNUH003-04)

Parents thought that having materials, such as posters, leaflets, and videos to communicate facts and figures about the impacts of smoke on their baby, would be effective, especially if they also included hard-hitting images to reinforce the message from the baby’s perspective:


*“I’ll be honest, rather than told, I’d have to see it. I’m more of a visual kind of person. If I saw they’d done research and show your facts and figures and numbers on the computer where they actually done it and how they done it. And that was harmful, yeah, I’d give up smoking tomorrow.”*
(S NNUH03-05)

Participants suggested that opportunities to reinforce smoke-free messages and offer support could be incorporated into discussion about using oxygen, transitioning home, coping mechanisms, and breastfeeding:


*“I think we should be thinking about discharge planning for babies from the day that they are admitted in many different ways. Teaching parents on tube feedings and you know care for their baby right from the get-go and thinking about it. So for me it would be wrapping it into that rather than, you know, as part of a package of how do we make sure your house is baby ready or premature baby ready?”*
(FG 13.10)

#### 4.3.3. ACT—Offer Parents Support to Create a Smoke-Free Environment for Baby

HCPs in particular were keen to offer support to parents framed as promoting and enabling smoke-free environments. They felt that this would be less confrontational and more acceptable to parents than an intervention focused on quitting:


*“Lots of our parents find that this is a great moment to think about how to bring up the baby in the most healthy way. Breastfeeding is a brilliant option. Also smoke free homes is a brilliant option. Would you like some support for this for you or any members of your family? Rather than ‘do you smoke?’—which is what I would have definitely asked because like you say it was a red book thing. Whereas that will always draw a negative response really.”*
(FG 13.10)

HCPs felt that the intervention, in addition to offering alternatives, should also take action by promoting a harm reduction approach to those who could not or did not want to fully quit smoking, and advise on methods to minimise their baby coming into contact with smoke:


*“You’re offering a service to help people stop if they choose to, but not everybody’s going to choose to or feel ready to. So I think maybe you should offer like ways that you can at least minimise the amount of smoke in your home. You know, so whether that is [nicotine] chewing gum or making people aware that they should wear a different coat and that’s like only for going outside. That they wash their hands thoroughly after coming in.”*
(S NNUH14-15)

##### Offer Nicotine Alternatives and Other Support

There was strong consensus across both HCPs and parents that smokers should be signposted to, or even offered, alternatives to smoking whilst on the unit to promote a smoke-free NICU. Some also felt that this could empower parents and, could ultimately, potentially promote quitting.


*“I think anything that makes them not spend that time outside smoking is a positive but if it’s an e-cigarette, as long as the NHS deems that to be safe as an alternative then I think that’s fine.”*
(UHL 007)

A couple of parents commented that nicotine replacement therapy (NRT) would enable parents to spend more time with their baby as they would not have to go outside:


*“Yeah, because obviously you want to spend more time with your baby than pop out for a cigarette. Because that’s that craving in your mind … More bonding time with your child.”*
(S NNUH12-13)

Both parents and HCPs were concerned about cost, and HCPs highlighted that when it came to NRT they were not able to prescribe for parents as the baby was their patient. However, some HCPs mentioned that there was a precedent for offering NRT demonstrated in the antenatal care:


*“We’ve now changed our medicines policy so that midwives can prescribe nicotine replacement therapy to any pregnant in-patients, whilst they’re on the antenatal ward. And that’s so that we can promote the site as smoke free and then you know, there’s a chance there to educate women about the harms of smoking etc. So we also have devised a sheet as well that they have to sign, whether they accept the NRT. It also says that they won’t smoke during their admission here, and if they decline the NRT then it’s basically saying that we can’t take responsibility for them if they go off site to smoke.”*
(FG 28.09)

Some parents commented that e-cigarettes satisfied the need for a break from the NICU, which some parents needed due to the intense nature of the situation. However, HCPs worried that e-cigarettes could dilute the message about smoking round baby. Many HCPs and parents were uneasy about e-cigarettes and believed they were still an unproven cessation method with potential health risks.


*“*
*I think that there, um, I from what I understand about them, they’re almost as bad as cigarettes themselves. Um, they’ve… they’ve got a lot of—and I may be very wrong in this—but I understand that they’ve got a lot of things in them that are dangerous as well…”*



*“I think from a baby point of view it’s obviously safer for the baby, so for the purposes of, you know the study with the outcome of you know improving health outcomes for the baby. I’m sure it is a step in the right direction. I think as a medical professional I would struggle to recommend that somebody do that because for personal health, I think it’s a really bad idea.”*
(FG 14.10.20)

In addition to alternatives, parents discussed other digital support including aids such as timelines, goal setting, distraction games, motivational messages, real-time interactions with healthcare professionals and other quitters, and even endorsements from celebrities and influencers. However, participants also emphasised that digital support should be specific to the postnatal situation of having had a baby on the NICU. It was felt that this could be tailored to support parents, immediately post-discharge:


*“Yeah, if someone text me say tomorrow and say, ‘did you manage to do your first night?’. I’d quite happily reply ‘look I managed to do it but I’m pulling my hair out here.”*
(ES NS NNUH12-13)

##### Smoke-Free Environment Promotion

When it came to implementing a *smoke-free NICU*, many of the parents felt there should be expectation setting and rules so that every parent entering the NICU knew that they should try and avoid exposing their baby, and potentially others, to tobacco smoking particles. They felt that timed absences following a cigarette could be a good option with some even suggesting using CO monitoring to implement this. Changing clothes or wearing protective clothes were also suggested:


*“Please respect that space and try to refrain from smoking for x amount of time, however long it needs to be for the effects of the smoke to be minimised”.*
(NS NNUH01)

All parents stated that they could benefit from discussions around *smoke-free home environments*, as even non-smoking parents commented that they had smoking friends or family members who were likely to come into contact with their baby. However, HCPs discussed that extended family members could be hard to engage with, and parents expressed concern that they did not know how to talk to their smoking relatives about the importance of keeping the home smoke free. One aspect of a proposed intervention included having leaflets to share with family members, and stickers and signs to put up at home to inform visitors that the house was smoke free. Some parents discussed that having information to share with smoking relatives could be empowering and help enable them to become smoke-free advocates for their baby. It was important that they could say it had been given to them by the hospital:


*“Some people might find it hard to explain to grandparents or mums or dads or brothers or sisters about how they want us to make smoke-free homes. How it can affect the baby and stuff like that. But if they can see it in black and white themselves then, you know, I think it might be a bit better. And make the mum feel less guilty on having to tell them.”*
(S NNUH16)

Although a couple of parents were worried that the stickers and signs could single out certain family members, most parents particularly liked the stickers and signs because they felt that they could avoid confrontation by showing a clear ‘official’ message:


*“Yeah. Because I think when, especially if you’ve got a premature baby, I think it’s a good … It’s a way of kind of avoiding an uncomfortable situation for you. Telling your friend ‘cause if your friend is going to walk in having smoked, she may think oh wait, hang on.”*
(NS UHL 020 21)

### 4.4. Intervention Logic Model and Diagram

The intervention, as defined by the logic model, background theory, and qualitative feedback presented in this paper, is a bespoke package of resources to be used by NICUs in the implementation of the NHS long term plan [22] (see Figure 2).

## 5. Discussion

Qualitative data gathered from mothers, fathers, a partner who was not a parent, family members and HCPs caring for babies admitted to NICUs indicated a clear consensus of the need for support to enable families to maintain smoke-free homes. Most parents considered it feasible and acceptable to ask families about tobacco smoking in the home from the time of admission of the baby. Many parents expected to be asked about smoking status and were surprised that this had never been asked. They were also amenable to CO testing as a way of ascertaining smoking status providing it was handled sensitively and offered to all parents, so as to avoid stigmatising those that smoke. Overall, it was felt that it would be acceptable, and indeed there was strong support for a clearly ‘smoke-free NICU’, where tobacco use during the time of the baby’s admission was not acceptable at all. Key to this was addressing the culture on NICU and enabling HCPs to feel informed and able to have conversations with families about the importance of maintaining a smoke-free home as part of their usual day-to-day practice.

For parents who smoke, support for a smoke-free home might involve direct support during the baby’s NICU admission to quit smoking. Offering alternatives including NRT and potentially e-cigarettes was supported. This was felt to be a positive way in which parents could be encouraged to avoid smoking completely during the NICU admission, which might mean that parents then went on to remain smoke-free. It was also well recognised that for ex-smokers, the NICU admission was a stressful time and there was a need to encourage recent ex-smokers to stay smoke-free and avoid relapse. However support for promotion of e-cigarettes was mixed, as there was some misgiving and confusion apparent in the data, despite Public Health and professional body guidance supporting the use of e-cigarettes for people trying to quit smoking [34]. Even for non-smokers, support was felt important to emphasise the importance of maintaining a smoke-free home environment for the baby.

There was strong consensus that the best way to educate parents about smoke free homes would be to take a ‘hard hitting’ approach to intervention. Health messages coming clearly from the perspective of the baby were felt to be the most impactful. However, the framing of these could be positive, i.e., the health benefits for the baby of staying smokee-free. These messages might be delivered in multi-media format—there was some support for posters and messaging on NICU. Leaflets were useful for some people but not others, and digital support was generally thought to be helpful, although not all parents engaged with apps and online forums. However, it is clear that moving towards a paper-free environment is critical for infection control reasons and, thus, parents described how they often spent extended periods of time on their phones while visiting their infant, and that this could present an ideal opportunity for education and support.

There was clear consensus for a continuity of care approach within the unit and beyond, such that all educational materials were supported and emphasised by NICU staff. Nurses and doctors might be trained to deliver basic smoke-free home advice, and this could be supplemented through referral to a specialist smoking cessation advisor based on the unit. Continuity was also critical across the time of the NICU admission and discharge—so support starting right from the admission should be reiterated throughout the in-patient stay, and then re-addressed in the community by an outreach nurse or a health visitor. Finally, intervention components that act as environmental prompts in the home, such as stickers and leaflets for family and friends, were thought to be potentially important aspects of an intervention that must also reach into the home environment in a sensitive way.

Previous studies have tested approaches to supporting smoke-free homes in the NICU setting. These have included education and motivational interviewing [35], but to date, no specific tailored approach has been developed for use in the context of UK NICUs. This study is the first step in meeting this need, in order to address smoking in families of extremely vulnerable and ‘at risk’ infants. We propose a targeted intervention approach, theoretically underpinned and specified via a logic model, demonstrating a pathway of support throughout the NICU admission and beyond. Support is clear and educational messages are ‘hard-hitting’ but delivered sensitively by trained NICU staff and positively offering harm reduction alternatives to promote a completely smoke-free NICU.

Findings of this study are limited in that the qualitative data were collected specifically for purposes of developing an intervention. Sampling was purposive rather than to achieve statistical generalisability. Analysis took a thematic approach, specifically coding for views and experiences to contribute towards the goal of intervention development. However, we gathered a range of views across two major NICUs. Positively, we managed to engage both fathers as well as mothers during the admission, capturing views from parents of a range of ethnicities, and interviewing both people who smoked as well as non-smokers. We also sampled across the wide range of HCPs who routinely delivered patient care in NICUs, and took a theoretically informed approach, guided by our logic model.

## 6. Conclusions

Following the detailed qualitative data gathered through this study, it is clear that a package of support should be implemented to provide the intervention that families of babies admitted to NICU want and need to maintain smoke-free homes. As this is a clear need, and it is recommended by current policy [22], it was felt that a further trial before implementation is not warranted. Qualitative findings reported support of a brief intervention ‘3As’ approach that should be implemented, where parents are asked about smoking status on admission of their baby to NICU, are advised of the health risks to their baby of environmental tobacco smoke exposure, and are actively supported to move away from tobacco, to maintain smoking abstinence, and to promote smoke-free homes to visitors, as appropriate. This is a novel approach and is the first UK intervention approach developed to create smokefree NICUs.

## Figures and Tables

**Figure 1 ijerph-19-03670-f001:**
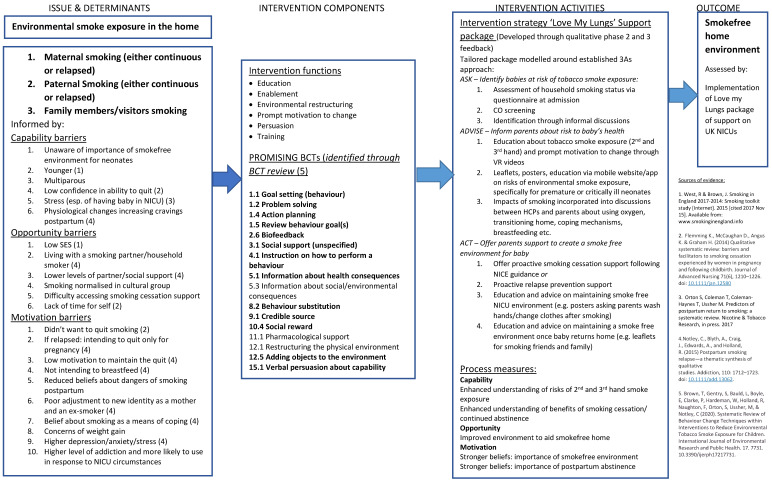
Intervention logic model.

**Figure 2 ijerph-19-03670-f002:**
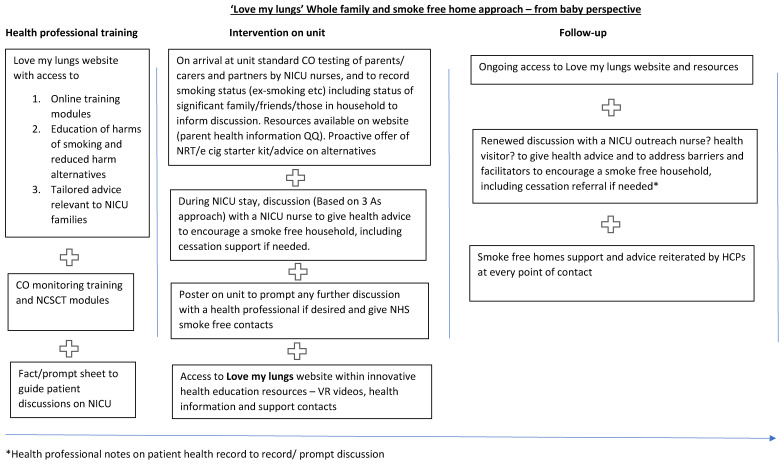
Intervention pathway.

**Table 1 ijerph-19-03670-t001:** Demographics of the interview sample.

	n
Gender	
Female	24 (57%)
Male	18 (43%)
Non-binary	0
Ethnicity	
White	33 (79%)
Asian/Asian British	7 (17%)
Mixed/multiple ethnic groups	1 (2%)
Black/African/Caribbean/Black British	1 (2%)
Highest Level Qualification	
None	5 (12%)
GCSE or equivalent	10 (24%)
A level or equivalent	11 (26%)
Further education	8 (19%)
University degree or above	8 (19%)
Relationship to baby admitted to NICU	
Mother	22 (52%)
Father	18 (43%)
Partner of mother/father (not biologically related to child)	1 (2%)
Grandparent	1 (2%)
Age	
Age range (years)	23–45
Mean age	33
Smoking status	
Current smoker (smoke one or more tobacco cigarettes per day)	10 (24%)
Recent ex-smoker (quit smoking tobacco in the last 12 months)	2 (5%)
Long-term ex-smoker (quit smoking tobacco completely more than 12 months ago)	6 (14%)
Experimented with tobacco smoking when younger but never smoked regularly	9 (21%)
Never smoked tobacco	15 (36%)

**Table 2 ijerph-19-03670-t002:** Overview of HCPs sample.

UHL	Play specialist, n = 1
	Advanced neonatal nurse practitioner (ANNP), n = 2
	Homecare nurse, n = 2
	Consultant neonatologist, n = 2
	Specialist trainee in paediatrics n = 2
NNUH	ANNP
	Senior sister, n = 2
	Outreach sister, n = 1
	Staff nurse, n = 2
	Nursery nurse, n = 2
	Senior clinical fellow, n = 1
	Matron, n = 1
	Smoking cessation midwife, n = 1
	Family care sister, n = 1
	Consultant neonatologist, n = 1
	NICU sister, n = 1
	Health visitor (community based), n = 1

## Data Availability

Data are available on request.

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
