# Peer review of "Development of a Smoke-Free Home Intervention for Families of Babies Admitted to Neonatal Intensive Care"

_ijerph, 2022, doi:10.3390/ijerph19063670_

Round 1

Reviewer 1 Report

See attached comments.

Reviewer 2 Report

Thank you very much for allowing me to review the article entitled “Development of a smoke-free home intervention for families of babies admitted to neonatal intensive care” (ijerph-1599192). Submitted for Special Issue “Smoking Cessation in Pregnancy”. This is a qualitative study, they used qualitative methods to develop an intervention that is relevant to the population it targets, is acceptable, feasible to implement, and therefore likely to be effective, cost effective and sustainable for promoting smoke-free home environments. the article is very well written. The title is appropriate to the content of the article and is therefore informative. The summary is well structured and informative Introduction: it is very well written, it is concise, it raises the working hypothesis and the bibliography used is adequate. Material and methods: presents the approval of the ethics committee. The inclusion and exclusion criteria should be explained. How much was met by the 1,170 infants who received Neonatal Intensive Care Unit care in 2018 and the participation rate of the population participating in the study. I suggest the realization of a flow chart that allows to see the characteristics of the population that finally participates with respect to the population served. I also suggest the realization of a scheme of how the intervention is carried out by health professionals. Result: In line 199 in which those fathers and mothers whose baby had died are eliminated, this information should be in the methodology and not in the result. as a result, the number of them should be put. The results are presented as a qualitative study stating the main responses of the participants. I think part of the results could be summarized. Figure 1 and two on line 619 and 621 are not accessible. this figure should be added in order to be evaluated. The differences between the situation before and after the intervention should be clearly presented and the achievements achieved by it. Discussion: it is very well written but I think it would need more bibliography that could shed light on the results obtained and the proposal presented. Conclusion: the conclusion must be the contribution of the experience presented and not a summary of the work.

Round 2

Reviewer 1 Report

See PDF.

Reviewer 2 Report

I have carefully reviewed the manuscript, again, entitled “Development of a smoke-free home intervention for families of babies admitted to neonatal intensive care” (ijerph-1599192). Submitted for Special Issue “Smoking Cessation in Pregnancy”.

 I don't understand why we can't see figure 1 and two that had been requested.

The authors have not increased the discussion as well as their bibliographic references and on the other hand, as indicated, in the conclusions bibliography cannot be used but rather a summary of the findings must be presented and adjusted to the limitations of the sample size on which they work, I suggest considering the need for future studies based on the orientation of their results.